

# Development and internal validation of a nine-lncRNA prognostic signature for prediction of overall survival in colorectal cancer patients

Zhiqiao Zhang[1,*], Qingbo Liu[2,*], Peng Wang[1], Jing Li[1], Tingshan He[1], Yanling Ouyang[1], Yiyan Huang[1] and Weidong Wang[2]

[1] Department of Infectious Diseases, Shunde Hospital, Southern Medical University, Shunde District, Guangdong, China, Shunde, Guangdong, China
[2] Department of Hepatobiliary Surgery, Shunde Hospital, Southern Medical University, Shunde, Guangdong, China
[*] These authors contributed equally to this work.

## ABSTRACT

**Background**. Colorectal cancer remains a serious public health problem due to the poor prognosis. In the present study, we attempted to develop and validate a prognostic signature to predict the individual mortality risk in colorectal cancer patients.

**Materials and Methods**. The original study datasets were downloaded from The Cancer Genome Atlas database. The present study finally included 424 colorectal cancer patients with wholly gene expression information and overall survival information.

**Results**. A nine-lncRNA prognostic signature was built through univariate and multivariate Cox proportional regression model. Time-dependent receiver operating characteristic curves in model cohort demonstrated that the Harrell's concordance indexes of nine-lncRNA prognostic signature were 0.768 (95% CI [0.717–0.819]), 0.778 (95% CI [0.727–0.829]) and 0.870 (95% CI [0.819–0.921]) for 1-year, 3-year and 5-year overall survival respectively. In validation cohort, the Harrell's concordance indexes of nine-lncRNA prognostic signature were 0.761 (95% CI [0.710–0.812]), 0.801 (95% CI [0.750–0.852]) and 0.883 (95% CI [0.832–0.934]) for 1-year, 3-year and 5-year overall survival respectively. According to the median of nine-lncRNA prognostic signature score in model cohort, 424 CRC patients could be stratified into high risk group ($n = 212$) and low risk group ($n = 212$). Kaplan–Meier survival curves showed that the overall survival rate of high risk group was significantly lower than that of low risk group ($P < 0.001$).

**Discussion**. The present study developed and validated a nine-lncRNA prognostic signature for individual mortality risk assessment in colorectal cancer patients. This nine-lncRNA prognostic signature is helpful to evaluate the individual mortality risk and to improve the decision making of individualized treatments in colorectal cancer patients.

Corresponding author
Peng Wang, sdgrxjbk@smu.edu.cn, wangpeng1962@yeah.net

## INTRODUCTION

Colorectal cancer (CRC) is one of the most common malignant tumors and one of the leading causes of cancer-related death, resulting in 135,430 estimated new cases and 50,260 estimated deaths in the United States in 2017 (*Siegel et al., 2017*; *Siegel, Miller & Jemal, 2016*). The 5-year survival rates of CRC patients varied dramatically in patients with different tumor stages (*Ferlay et al., 2010*). The 5-year survival rate of localized CRC patients was 90%, whereas it was 71% and 14% for CRC patients with regional metastasis and distant metastasis respectively (*Siegel et al., 2017*). It was reported that the 5-year survival rate was approximately 10% in stage IV CRC patients (*Sanoff et al., 2008*). Therefore, it is of great importance to develop a reliable prognostic biomarker to predict the prognosis and optimize the clinical therapy decision.

Long non-coding RNAs (lncRNAs) are a class of non-coding RNAs more than 200 nucleotides in length (*McFadden & Hargrove, 2016*). The lncRNAs have been reported to be correlated with overall survival and might serve as prognostic biomarkers for CRC patients (*Kita et al., 2017*; *Weng et al., 2017*; *Xie et al., 2016*). Recently, several studies constructed lncRNAs-based prognostic signatures to predict the overall survival of CRC patients by using Cox proportional regression model (*Fan & Liu, 2018*; *Xing et al., 2018*; *Xue et al., 2017*; *Zeng et al., 2017*). However, these prognostic signatures have three limitations for clinical application. First, these prognostic signatures are too difficult to calculate for clinical application due to the complex formulas. For example, the prognostic signature constructed by *Xing et al. (2018)* did not provide detailed formula of random forest model and therefore was unrepeatable for clinical application. The study performed by Xue et al. calculated the prognostic risk score by using reads per kilobase per million mapped reads (RPKM) method. However, as the original gene expression read counts were not available for most studies, RPKM method was difficult to perform for clinical application in different population (*Xue et al., 2017*). Second, the clinical significances of prognostic risk scores in theses previous studies were difficult to understand for patients without medical knowledge. Third, these prognostic signatures provided merely the prediction of overall survival for a special subgroup, but not the individual risk prediction of overall survival. Therefore, an ideal individual risk predictive model should be easy to obtain, calculate and understand for clinical application.

Several studies have used nomogram method to predict the prognosis of different cancers (*Li et al., 2015*; *Tian et al., 2017*). This method has two advantages in predicting the prognosis. Firstly, the nomogram is convenient to calculate and evaluate the individual probability of disease without complex formula. Secondly, this method provides a straightforward individual risk and the result is easy to understand for patients. Therefore, the nomogram method is suitable for prediction of prognosis. To the best of our knowledge, the present study is the first to develop a prognostic nomogram by using the lncRNA expression data for overall survival of CRC patients.

In the present study, our objective was to develop and validate a lncRNA-based prognostic signature to predict the overall survival of CRC patients. To improve the quality of prognostic model, the development and validation of the prognostic model

in the present study were performed in accordance with the guidelines of Transparent Reporting of a multivariable prediction model for Individual Prognosis Or Diagnosis (TRIPOD) (*Collins et al., 2015*).

## MATERIALS AND METHODS

### Study protocol approval

The data download and analyses were performed according to the policies of The Cancer Genome Atlas (TCGA) database. Since the study datasets in the present study were all downloaded from TCGA database, additional ethics approval was not needed. All data collections and analyses were in accordance with the principles of Declaration of Helsinki.

### The gene expression dataset

The original RNA sequencing dataset was obtained from GDC Data Portal (https://portal.gdc.cancer.gov/exploration). The dataset was downloaded in May 31, 2018. The RNA sequencing data were generated on the Illumina HiSeq 2000 RNA Sequencing platform. The downloaded RNA sequencing data contained the original gene read counts of 60,483 genes from 458 CRC tumor tissues and 41 non-tumor normal tissues. The duplicated samples were removed from the present study ($n = 22$). In the present study, the lncRNAs ID were downloaded from GENCODE Version 7 (*Derrien et al., 2012*). Finally the extracted gene expression data of 14,449 lncRNAs from 458 CRC tissues and 41 non-tumor normal tissues were selected for differentially expressed lncRNAs.

### Differential expression analyses

The "edgeR" package was applied for the differential expression analyses and the original gene expression read counts was normalized by the Trimmed Mean of M (TMM) method (*Robinson & Oshlack, 2010*). The genes whose average expression read counts lower than 1 were filtered out from the present study. The *F*-tests was used for assessments of quasi-likelihood. Thresholds of differentially expressed lncRNAs were *P* adj <0.05 and |log2 fold change| >2.

### Heat map and volcano map

The heat map and volcano map were drawn according to the normalized gene expression values of differentially expressed lncRNAs. The darker color represented higher expression level of differentially expressed lncRNA.

### The model cohort

The clinical dataset of 629 CRC patients were downloaded from cBioPortal database (May 31, 2018, http://www.cbioportal.org/datasets). The patients with overall survival time less than one month were excluded from the present study to avoid the impact of unrelated causes of death ($n = 49$). The clinical dataset was matched to the gene expression dataset. The patients without gene expression information were excluded from the present study ($n = 156$). Finally 424 CRC patients were enrolled for further survival analysis. Figure 1 was the study flowchart for patient selection in the present study. The end-point in the present study was overall survival (OS). The average follow-up time was 30.0 ± 25.5
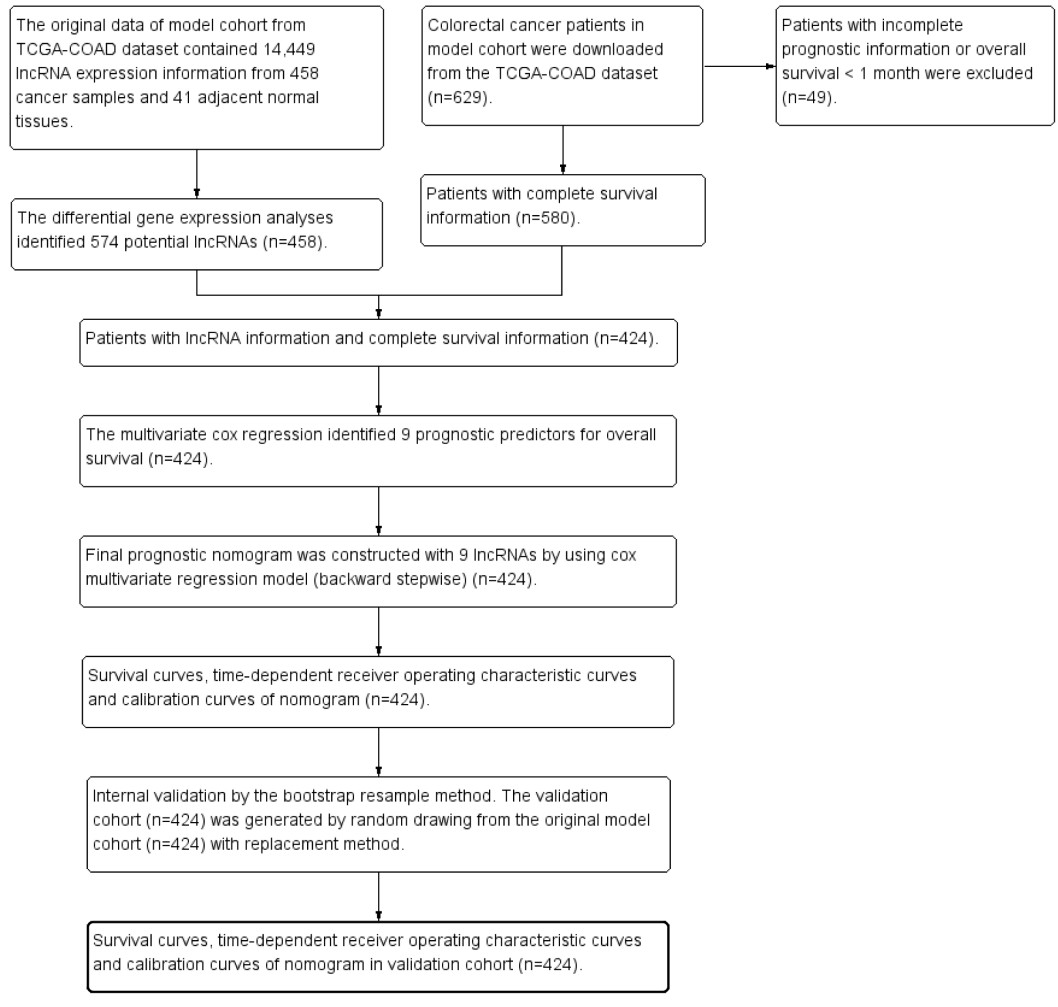

**Figure 1** **Study flowchart in the present study.** TCGA, The Cancer Genome Atlas

months. Overall survival time was calculated from initial diagnosis time to death time or the last follow-up time. The maximum value and the minimum value of overall survival time were 140.9 months and 1.0 month. The study time was from May 18, 2010 to January 27, 2015. The missing data in the present study were coded as "NA" in all tables and analyses.

## Assessment of predictive performance

The Harrell's concordance index (C-index), calibration plot and time-dependent receiver operating characteristic (ROC) curves were used to assess the predictive performance of prognostic signatures. The predictive performance of prognostic signature in the current study was compared with two previous prognostic signatures (named RS lncRNA score and Risk score). The formulas of the RS lncRNA score and the Risk score were as follows:

RS lncRNA score $= (0.1337 \times$ expression value of RP1-170O19.17$) + (0.0633 \times$ expression value of RP11-785D18.3$) + (0.8131 \times$ expression value of RP11-798K3.2$)$

$+ (-1.5194 \times$ expression value of XXbac-B476C20.9) $+ (4.3132 \times$ expression value of RP11-481J13.1) $+ (0.4497 \times$ expression value of RP11-167H9.4) (*Fan & Liu, 2018*). In the formula of RS lncRNA score, the lncRNA expression values were normalized by "DEseq" packages.

Risk score $= \exp_{LINC01555} * (-.191) + \exp_{RP11-108K3.1} * (0.318) + \exp_{LINC01207} * (-0.191) + \exp_{RP11-610P16.1} * (-0.163)$ (*Zeng et al., 2017*). In the formula of Risk score, the lncRNA expression values were $\log_2$-transformed.

## Internal validation by using bootstrap resampling method

The bootstrap resampling method has been recommended for internal validation of the predictive model (*Blackstone, 2001*; *Grunkemeier & Wu, 2004*). The validation cohort in the present study was constructed by drawing 424 CRC patients with replacements from the original model cohort.

## Statistical analysis

The statistical analyses were carried out by using the R software (version 3.4.1) and SPSS Statistics 19.0 (SPSS Inc., Armonk, NY, USA). The following R packages were carried out as needed in the current study: "survival", "rms" , "pROC", "plyr", "glmnet", and "timeROC". Continuous variables in the present study were presented as mean $\pm$ standard deviation. Continuous variables were compared by $t$-test or Mann–Whitney $U$ test. Categorical variables were compared by chi-squared test or Fisher's exact test. A 2-tail $P < 0.05$ was defined as statistically significant in the present study.

# RESULTS

## Study cohorts

The present study finally included 424 CRC patients with total lncRNA expression information and overall survival information. The average age was $66.7 \pm 13.0$ years and the average overall survival time was $30.0 \pm 25.5$ months in model cohort. There were 102 (24.0%) patients died during the follow-up period in model cohort, whereas there were 108 (25.5%) patients died during the follow-up period in validation cohort. The clinical characteristics of CRC patients in model cohort and validation cohort were summarized in Table 1. There were no significant differences in terms of clinical characteristics and lncRNA expression between model cohort and validation cohort. There were no missing data in terms of survival status, survival time and lncRNA expression value.

## Differentially expressed analyses

We performed differential expression analyses by comparing all gene expression values between 458 tumor tissues and 41 normal colon tissues. The "edgeR" package identified 574 differentially expressed genes for overall survival. The heat map and volcano map of differential expression genes were presented in Figs. S1 and S2, respectively.

## Construction of prognostic signature for overall survival

The univariate Cox regression analysis was performed to explore the potential prognostic lncRNAs for overall survival. The univariate Cox proportional regression analysis

**Table 1  The clinical features of colorectal cancer patients in model cohort and validation cohort.**

|  | Model cohort ($n = 424$) | Validation cohort ($n = 424$) | P value |
|---|---|---|---|
| Death (n(%)) | 102(24.0) | 108(25.5) | 0.633 |
| Survival time (mean $\pm$ SD, month) | 30.0 $\pm$ 25.5 | 28.6 $\pm$ 25.4 | 0.163 |
| Age (mean $\pm$ SD, year) | 66.7 $\pm$ 13.0 | 67.1 $\pm$ 12.4 | 0.612 |
| Gender (Male/Female) | 230/194 | 224/200 | 0.680 |
| Tumor site |  |  |  |
| Colon | 361(85.1) | 351(82.8) | 0.363 |
| Rectum | 58(13.7) | 69(16.3) |  |
| NR | 5(1.2) | 4(0.9) |  |
| AJCC Stage (IV/III/II/I/NA) | 59/124/162/68/11 | 58/145/143/68/10 | 0.578 |
| AJCC PT (T4/T3/T2/T1/NA) | 51/292/70/11/0 | 54/286/73/11/0 | 0.976 |
| AJCC PN (N2/N1/N0/NA) | 77/102/245/0 | 88/105/231/0 | 0.756 |
| AJCC PM (MX/M1/M0/NA) | 47/59/312/6 | 46/58/314/6 | 0.998 |
| Radiation treatment adjuvant (Yes/No /NA) | 0/33/391 | 0/42/382 | 0.553 |
| Pharmaceutical adjuvant (Yes/No /NA) | 18/15/391 | 22/20/382 | 0.608 |
| History other malignance (Yes/No /NA) | 56/368/0 | 50/374/0 | 0.533 |
| AC005256.1 (High/Low) | 212/212 | 212/212 | 1.0 |
| RP11_815M8.1 (High/Low) | 212/212 | 219/205 | 0.631 |
| RP11_342A23.2 (High/Low) | 212/212 | 212/212 | 1.0 |
| RP11_264B14.1 (High/Low) | 212/212 | 196/228 | 0.272 |
| AC064834.1 (High/Low) | 212/212 | 215/209 | 0.837 |
| RP11_108K3.2 (High/Low) | 212/212 | 207/217 | 0.731 |
| LINC01571 (High/Low) | 212/212 | 206/218 | 0.680 |
| RP11_383I23.2 (High/Low) | 212/212 | 206/218 | 0.680 |
| AC079612.1 (High/Low) | 212/212 | 237/187 | 0.085 |

**Notes.**

Continuous variables were compared by $t$-test or Mann–Whitney $U$ test as appropriate. Categorical variables were compared by chi-squared test or Fisher's exact test as appropriate.

NA, missing data; NR, not reported.

identified 33 potential lncRNA predictors for overall survival. Using multivariate Cox proportional regression analysis, a nine-lncRNA prognostic signature (Fig. 2) was constructed based on the potential prognostic lncRNA predictors determined by Cox regression analysis. The overall information of nine prognostic lncRNA predictors were summarized in Table 2. The formula of nine-lncRNA prognostic signature was as follows: nine-lncRNA prognostic signature score = (0.837* RP11_815M8.1) + (0.822* RP11_342A23.2) + (0.905* RP11_264B14.1) + (−0.529* AC064834.1) + (0.907* RP11_108K3.2) + (−0.745* LINC01571) + (1.241* RP11_383I23.2) + (−0.737* AC079612.1) + (0.725* AC005256.1).

## Performance of nine-lncRNA prognostic signature in model cohort

The nine-lncRNA prognostic signature score were generated according to the previous formula. The distributions of nine-lncRNA prognostic signature score (Fig. 3A), overall survival status and overall survival time (Fig. 3B) in model cohort were shown in Fig. 3.

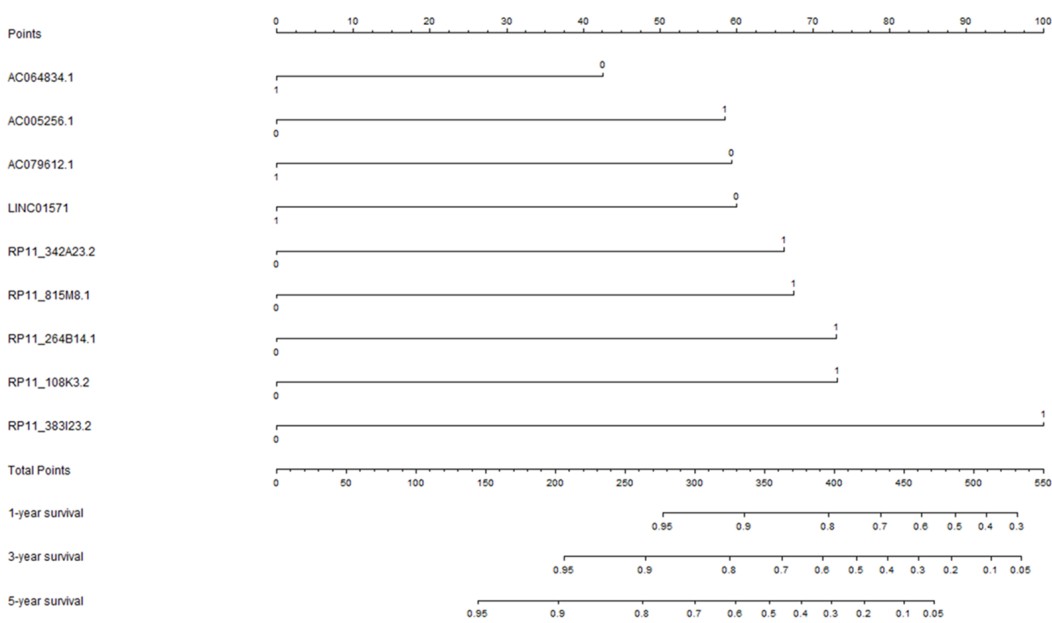

**Figure 2  The nine-lncRNA prognostic signature to predict the overall survival of colorectal cancer patients.**

**Table 2  The overall information of nine prognostic lncRNA predictors in univariate and multivariable Cox regression analyses.**

| | Univariate analysis | | | Multivariate analysis | | | |
|---|---|---|---|---|---|---|---|
| | HR | 95% CI | *P*-value | Coefficient | HR | 95% CI | *P*-value |
| RP11_815M8.1 (High/Low) | 1.842 | 1.232–2.754 | 0.003 | 0.837 | 2.310 | 1.473–3.623 | <0.001 |
| RP11_342A23.2 (High/Low) | 1.580 | 1.063–2.347 | 0.024 | 0.822 | 2.276 | 1.490–3.477 | <0.001 |
| RP11_264B14.1 (High/Low) | 1.510 | 1.015–2.246 | 0.042 | 0.905 | 2.471 | 1.582–3.857 | <0.001 |
| AC064834.1 (High/Low) | 0.590 | 0.396–0.880 | 0.010 | −0.529 | 0.589 | 0.391–0.889 | 0.012 |
| RP11_108K3.2 (High/Low) | 2.552 | 1.659–3.927 | <0.001 | 0.907 | 2.478 | 1.592–3.855 | <0.001 |
| LINC01571 (High/Low) | 0.661 | 0.446–0.981 | 0.040 | −0.745 | 0.475 | 0.307–0.733 | <0.001 |
| RP11_383I23.2 (High/Low) | 1.667 | 1.117–2.487 | 0.012 | 1.241 | 3.459 | 2.143–5.583 | <0.001 |
| AC079612.1 (High/Low) | 0.450 | 0.297–0.681 | <0.001 | −0.737 | 0.479 | 0.314–0.731 | <0.001 |
| AC005256.1 (High/Low) | 1.606 | 1.080–2.387 | 0.019 | 0.725 | 2.065 | 1.352–3.155 | <0.001 |

**Notes.**

HR, hazard ratio; CI, confidence interval.

The medians of lncRNA expression values were used as cutoff values to stratify lncRNA expression values into high expression group (as value 1) and low expression group (as value 0).

In model cohort, the Harrell's concordance-index (C-index) of nine-lncRNA prognostic signature was 0.757 (95% CI [0.706–0.808]).

## Kaplan–Meier survival curves and log-rank test in model cohort

The median of nine-lncRNA prognostic signature score was used as cutoff value to stratify CRC patients into high risk group and low risk group. Kaplan–Meier survival curves (Fig. 4) and log-rank test were used to compare the difference of overall survival rate

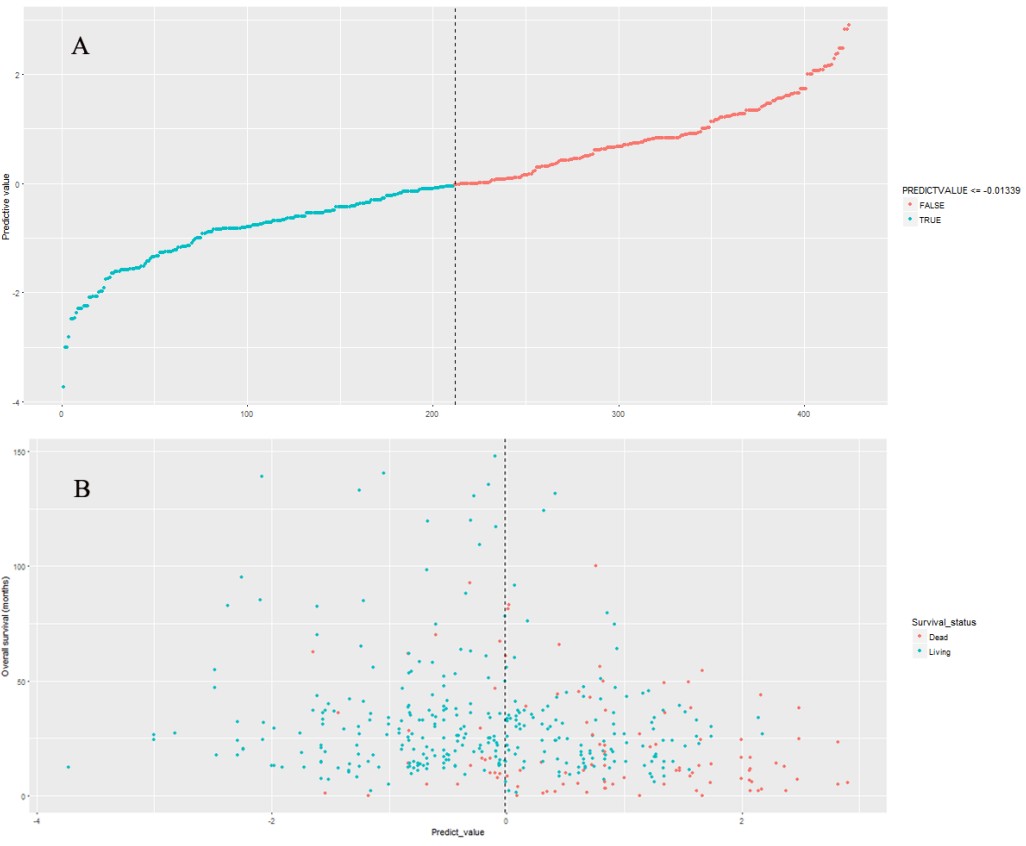

**Figure 3** The distributions of nine-lncRNA prognostic signature (A), overall survival status and overall survival time (B) in model cohort.

between high risk group and low risk group. As shown in Fig. 4, patients with high nine-lncRNA prognostic signature score had poorer overall survival rate than patients with low nine-lncRNA prognostic signature score ($P < 0.001$).

## Time-dependent receiver operating characteristic curves in model cohort

We further explored the predictive accuracy of nine-lncRNA prognostic signature score compared with two previous prognostic signatures by using time-dependent receiver operating characteristic curves (Fig. 5). The C-indexes of nine-lncRNA prognostic signature, RS lncRNA score and Risk score were 0.768, 0.654 and 0.658 for 1-year overall survival (Fig. 5A) respectively, whereas it were 0.778, 0.666 and 0.582 for 3-year overall survival (Fig. 5B). For 5-year overall survival (Fig. 5C), the C-indexes of nine-lncRNA prognostic signature, RS lncRNA score and Risk score were 0.870,0.681 and 0.633, respectively.

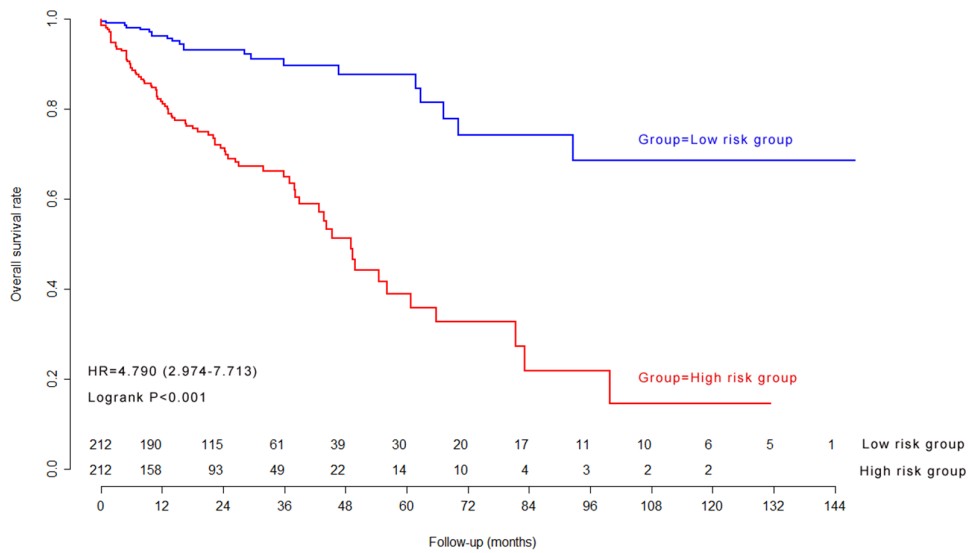

**Figure 4** **The Kaplan–Meier survival curves in model cohort.**

## Calibration curves in model cohort

The calibration curves were used to assess the predictive performance of nine-lncRNA prognostic signature. The calibration curves for 1-year (Fig. 5D), 3-year (Fig. 5E) and 5-year (Fig. 5F) overall survival demonstrated that there were a good agreement between the predictive probability and the actual overall survival in model cohort.

## Internal validation of nine-lncRNA prognostic signature

The validation cohort ($n = 424$) was generated by random drawing from the model cohort with replacement method. The nine-lncRNA prognostic signature score were generated according to the previous formula for patients in validation cohort. The distributions of nine-lncRNA prognostic signature score (Fig. 6A), overall survival status and overall survival time (Fig. 6B) in validation cohort were shown in Fig. 6. The C-index of nine-lncRNA prognostic signature was 0.751 95% CI [0.700–0.802]) in validation cohort.

## Kaplan–Meier survival curves and log-rank test in validation cohort

The previous cutoff value of nine-lncRNA prognostic signature score in model cohort was used as the cutoff value to stratify CRC patients into high risk group and low risk group in validation cohort. As shown in Fig. 7, the log-rank test indicated that the overall survival rate in high risk group was significantly lower than that in low risk group ($P < 0.001$).

## Time-dependent receiver operating characteristic curves in validation cohort

In validation cohort, the C-indexes of nine-lncRNA prognostic signature, RS lncRNA score and Risk score were 0.761, 0.695 and 0.664 for 1-year overall survival (Fig. 8A) respectively, whereas it were 0.801, 0.660 and 0.582 for 3-year overall survival (Fig. 8B).

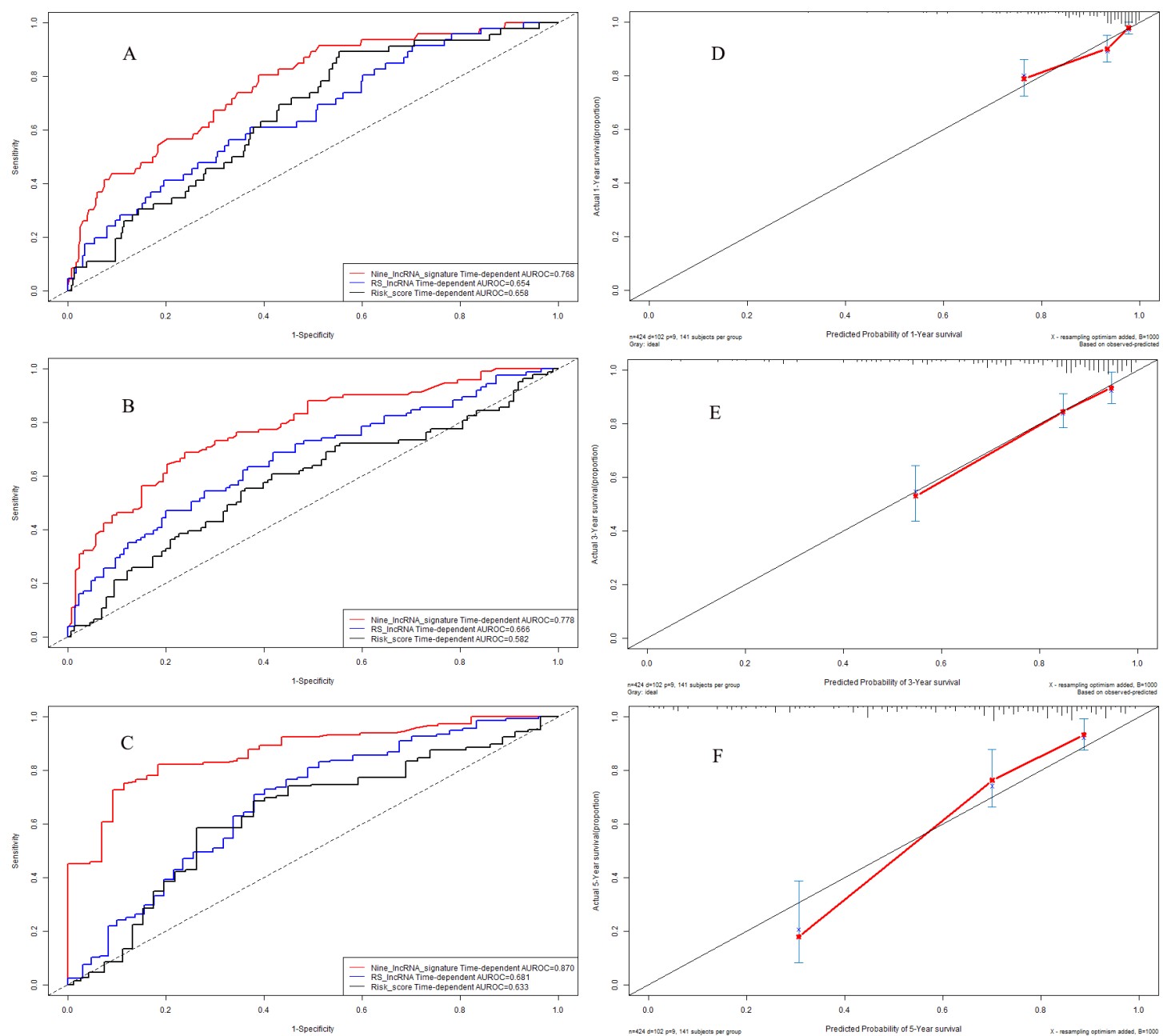

**Figure 5** Performance of nine-lncRNA prognostic signature in model cohort: Time-dependent receiver operating characteristic curves of three prognostic models according to 1-year (A), 3-year (B) and 5-year (C) overall survival. (D) Calibration curve for 1-year overall survival; (E) Calibration curve for 3-year overall survival; (F) Calibration curve for 5-year overall survival.

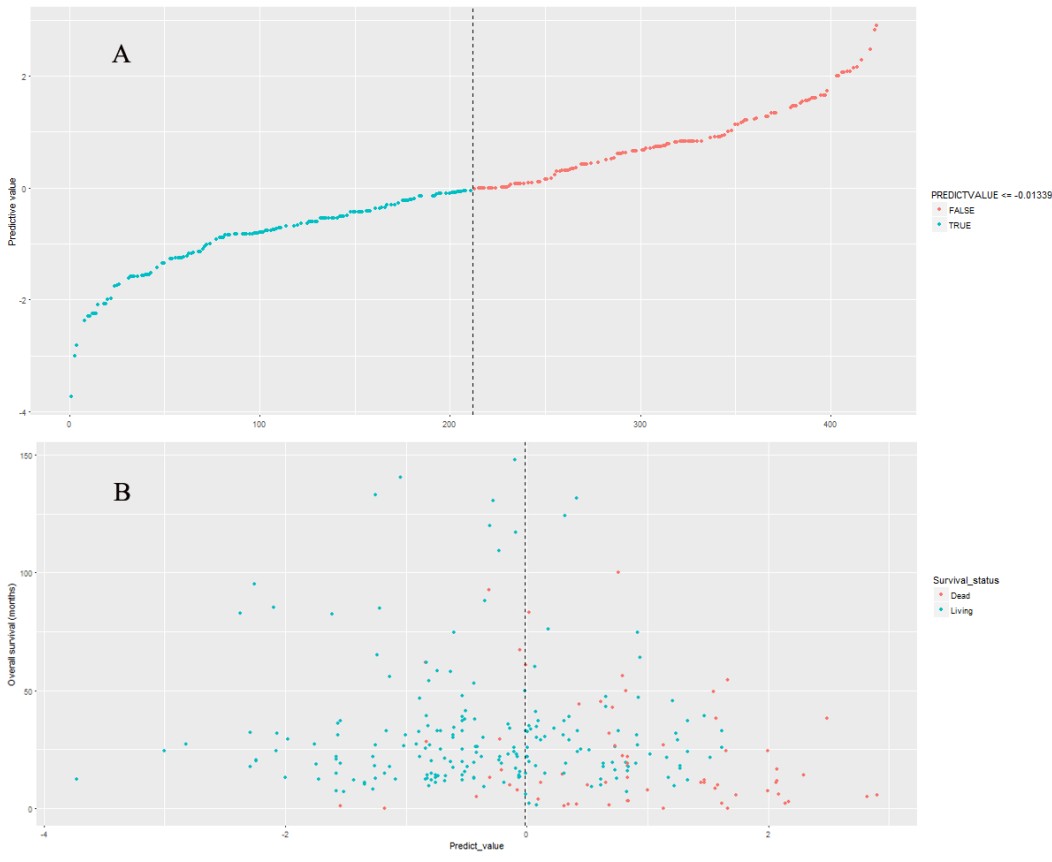

**Figure 6** The distributions of nine-lncRNA prognostic signature (A), overall survival status and overall survival time (B) in validation cohort.

For 5-year overall survival (Fig. 8C), the C-indexes of nine-lncRNA prognostic signature, RS lncRNA score and Risk score were 0.883,0.694 and 0.616 respectively.

## Calibration curves in validation cohort

The calibration curves for 1-year (Fig. 8D), 3-year (Fig. 8E) and 5-year (Fig. 8F) overall survival indicated a good agreement between the predictive probability of overall survival and the actual overall survival in validation cohort.

## Independence of nine-lncRNA prognostic signature for overall survival

We further carried out multivariate Cox regression analyses to explore whether nine-lncRNA prognostic signature was an independent influence factor for overall survival in model cohort and validation cohort. After adjustment of other clinical variables, including age, gender, the American Joint Committee on Cancer (AJCC) PT, AJCC PN, AJCC PM and AJCC stage (Table 3), the results of multivariate Cox regression analysis indicated that the nine-lncRNA prognostic signature was an independent influence factor for overall survival of CRC patients.

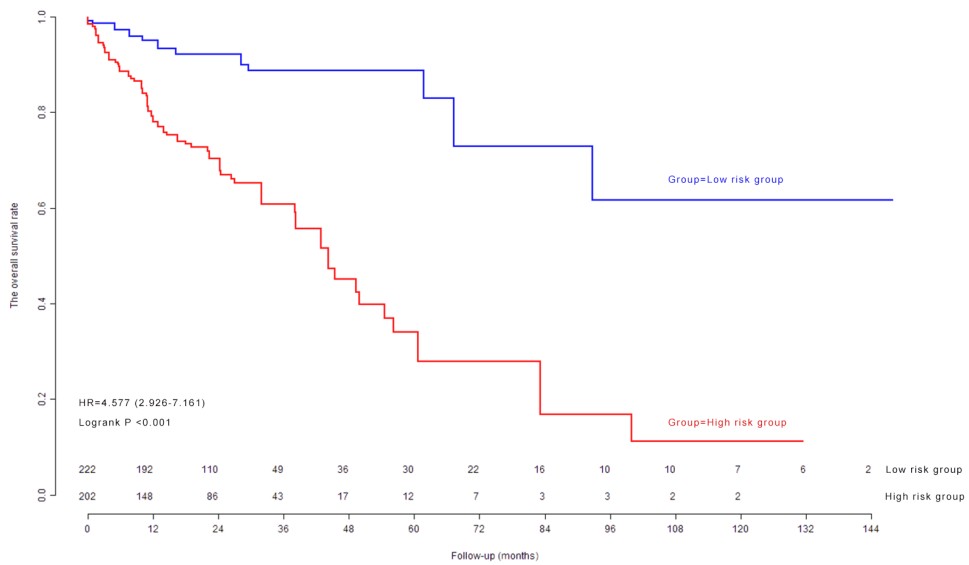

**Figure 7  The Kaplan–Meier survival curves in validation cohort.**

## Functional enrichment analysis of prognostic signature

We performed functional enrichment analysis to explore the biological pathway and process correlated with this nine-lncRNA prognostic signature. At first, the pearson correlation coefficients between these prognostic lncRNA expression values and the mRNA expression values were calculated in the TCGA dataset. Then the genes correlated with at least one of these prognostic lncRNAs (defined as |Pearson correlation coefficient| >0.5) were included into the following functional enrichment analysis. The gene ontology (GO) biological process enrichment analysis and the Kyoto Encyclopedia of Genes and Genomes (KEGG) signaling pathway analysis were presented in Fig. 9 by using the above identified genes in the Database for Annotation, Visualization, and Integrated Discovery (DAVID) (https://david.ncifcrf.gov/) Bioinformatics Resources. The results of functional enrichment analysis demonstrated that the co-expressed genes were mainly enriched in G-protein coupled receptor signaling pathway, potassium ion transmembrane transport, carboxylic acid metabolic process response to hormone, regulation of ion transmembrane transport (Fig. 9).

## DISCUSSION

In the present study, we developed and validated a nine-lncRNA prognostic signature that was helpful for individual mortality risk prediction and survival stratification of CRC patients. This nine-lncRNA prognostic signature was helpful for patients to ascertain their individual mortality risk and optimize their personalized treatment strategies. Time-dependent receiver operating characteristic curves demonstrated that this nine-lncRNA prognostic signature was superior to other two previous prognostic signatures for prediction of overall survival.

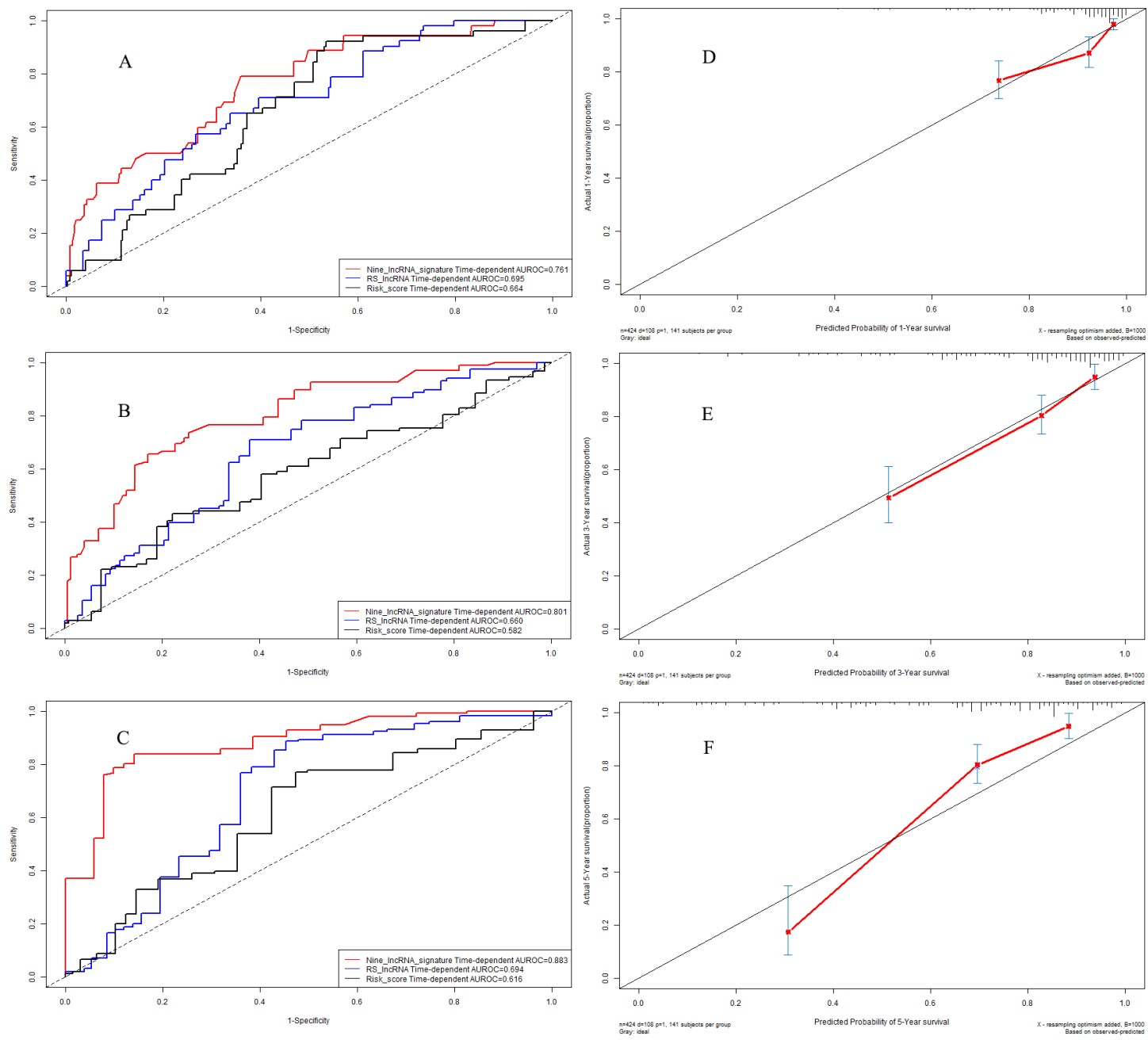

**Figure 8** Performance of nine-lncRNA prognostic signature in validation cohort: Time-dependent receiver operating characteristic curves of three prognostic models according to 1-year (A), 3-year (B) and 5-year (C) overall survival. (D) Calibration curve for 1-year overall survival; (E) Calibration curve for 3-year overall survival; (F) Calibration curve for 5-year overall survival.

This nine-lncRNA prognostic signature without pathological parameters provided a noninvasive preoperative prediction method for overall survival of CRC patients. Firstly, this nine-lncRNA prognostic signature could provide a preoperative individual mortality risk prediction, which was helpful for patients without medical knowledge to understand

**Table 3  Univariate and multivariable Cox regression analyses.**

| | Univariate analysis | | | | Multivariate analysis | | | |
|---|---|---|---|---|---|---|---|---|
| | *n* | HR | 95% CI | *P*-value | Coefficient | HR | 95% CI | *P*-value |
| **Model cohort (*n* = 424)** | | | | | | | | |
| Age(year) | 424 | 1.022 | 1.005–1.038 | 0.010 | 0.025 | 1.025 | 1.007–1.044 | 0.007 |
| Gender (Male/Female) | 424 | 1.112 | 0.752–1.645 | 0.595 | −0.101 | 0.904 | 0.595–1.373 | 0.636 |
| AJCC PT (T4,T3/T2,T1) | 424 | 2.926 | 1.356–6.318 | 0.006 | 1.161 | 3.193 | 1.248–8.165 | 0.015 |
| AJCC PN (N2,N1/N0) | 424 | 2.506 | 1.683–3.733 | <0.001 | −0.551 | 0.576 | 0.218–1.524 | 0.267 |
| AJCC PM (MX,M1/M0) | 418 | 2.962 | 1.985–4.419 | <0.001 | 0.707 | 2.029 | 1.284–3.205 | 0.002 |
| AJCC stage (IV,III/II,I) | 413 | 2.841 | 1.869–4.317 | <0.001 | 1.233 | 3.433 | 1.157–10.187 | 0.026 |
| Nine-lncRNA prognostic signature (High/Low) | 424 | 4.790 | 2.974–7.713 | <0.001 | 1.513 | 4.539 | 2.722–7.571 | <0.001 |
| **Validation cohort (*n* = 424)** | | | | | | | | |
| Age(year) | 424 | 1.017 | 1.002–1.033 | 0.032 | 0.031 | 1.031 | 1.012–1.050 | 0.001 |
| Gender (Male/Female) | 424 | 1.051 | 0.720–1.535 | 0.797 | −0.213 | 0.808 | 0.538–1.214 | 0.306 |
| AJCC PT (T4,T3/T2,T1) | 424 | 8.353 | 2.646-26.368 | <0.001 | 3.079 | 21.729 | 2.988–158.025 | 0.002 |
| AJCC PN (N2,N1/N0) | 424 | 2.253 | 1.528–3.323 | <0.001 | −0.526 | 0.591 | 0.260–1.346 | 0.211 |
| AJCC PM (MX,M1/M0) | 418 | 2.347 | 1.577–3.493 | <0.001 | 0.570 | 1.769 | 1.107–2.827 | 0.017 |
| AJCC stage (IV,III/II,I) | 414 | 2.741 | 1.817–4.134 | <0.001 | 1.231 | 3.425 | 1.317–8.905 | 0.012 |
| Nine-lncRNA prognostic signature (High/Low) | 424 | 4.577 | 2.926–7.161 | <0.001 | 1.515 | 4.551 | 2.806–7.383 | <0.001 |

Notes.

AJCC, the American Joint Committee on Cancer; HR, hazard ratio; CI, confidence interval.

The median of nine-lncRNA prognostic signature score was used as the cutoff value to stratify colorectal patients into high risk group and low risk group.

the actual individual mortality risk in different clinical endpoints. Secondly, the median of nine-lncRNA prognostic signature score was used as the cutoff value to stratify colorectal patients into high risk group and low risk group. The Kaplan–Meier survival curves demonstrated that the overall survival rate in high risk group was significantly lower than that in low risk group. Therefore, this nine-lncRNA prognostic signature was helpful for patients with high mortality risk to make clinical decision of receiving active individualized treatment.

As reported in the original article, the AUROCs of Risk score for 5-year overall survival were 0.706 and 0.619 in model group and validation group respectively (*Zeng et al., 2017*). The AUROCs of RS lncRNA score were 0.731 and 0.727 in training dataset and testing dataset (*Fan & Liu, 2018*). The AUROCs of Risk score and RS lncRNA score in the present study were lower than that in original articles. The differences of AUROC might be caused by the following reasons: firstly, the sample size in the present study was 424, whereas it were 371 for Risk score and 568 for RS lncRNA score. The difference of sample size would result in an influence to the predictive accuracy of prognostic models. Secondly, the numbers of selected lncRNAs were different in three prognostic signatures. The number of selected lncRNAs in nine-lncRNA prognostic signature was nine, whereas it was four for the Risk score and six for the RS lncRNA score. Thirdly, the Risk score was calculated by using the lncRNA expression values which have been $\log_2$-transformed. The RS lncRNA score was calculated by using the lncRNA expression values which have

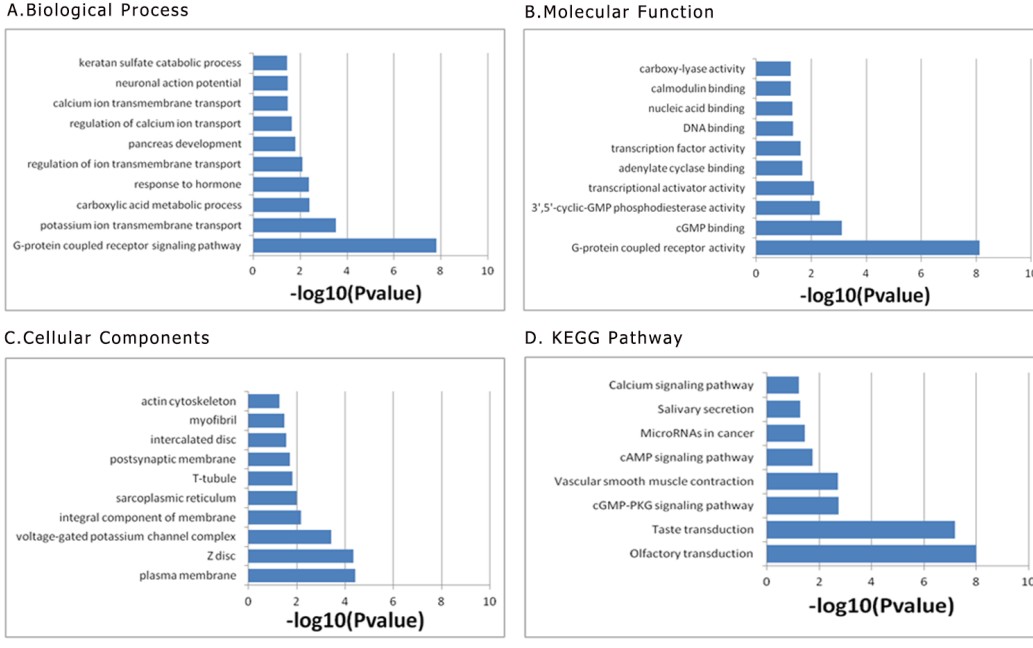

**Figure 9** Functional enrichment analysis of prognostic signature: (A) Biological Process; (B) Molecular Function; (C) Cellular Component; (D) KEGG Pathway. KEGG, Kyoto Encyclopedia of Genes and Genomes.

been normalized by the "DEseq" package. In order to improve the universal applicability of the research results, the medians of original lncRNA expression values were used as cutoff values to stratify lncRNA expression values into high expression group (as value 1) and low expression group (as value 0) in the present study. The nine-lncRNA prognostic signature score were calculated by using the binary lncRNA expression values according to the medians of original lncRNA expression values. This adjustment was helpful to improve the clinical application of prognostic model in other study population.

The present study has the following advantages: first, this nine-lncRNA prognostic signature can be used to assess the individual mortality risk through simple chart by patients. Second, this nine-lncRNA prognostic signature can provide an individual mortality risk assessment for 1-year, 3-year and 5-year overall survival. Individual mortality risk prediction at different time points is of great significance to persuade CRC patients receive a timely differentiated intensive treatment. Third, this nine-lncRNA prognostic signature provides individual mortality risk probability and the result is easy to understand for patients without medical knowledge. Fourth, this nine-lncRNA prognostic signature can serve as a preoperative non-invasive prediction method for overall survival of CRC patients. Therefore, this nine-lncRNA prognostic signature is suitable for preoperative prediction of overall survival, especially for advanced CRC patients who can't tolerate surgery.

The current study has two shortcomings which must be considered for interpreting the clinical significance of the results. First, the current study constructed a prognostic

signature by using nine lncRNA predictors. However, due to the different gene detection platforms and different lncRNA name, we could not obtain second independent dataset containing these nine lncRNA predictors in other databases including Gene Expression Omnibus (GEO) database, ArrayExpress database, and The Atlas of Noncoding RNAs in Cancer (TANRIC). We performed an internal validation by using bootstrap resampling method but not external validation by using independent data. Therefore, this nine-lncRNA prognostic signature in the current study need further external validation through external independent dataset. Second, we searched TCGA database and ascertained nine lncRNAs as prognostic predictors for overall survival in CRC patients. The relationship between these lncRNA predictors and the prognosis of CRC patients was not clear. Third, the average follow-up time was $30.0 \pm 25.5$ months and there were only 102 (24.0%) patients died during the follow-up period in model cohort. Taking into account the limited follow-up time and relatively small event size, the association between these lncRNA predictors and the prognosis of CRC patients should be validated in an additional cohort with longer follow-up period and larger event size. Therefore, large prospective clinical studies are needed to clarify the relationship between these lncRNA predictors and the prognosis of CRC patients.

## CONCLUSIONS

In summary, the present study developed and validated a nine-lncRNA prognostic signature for individual mortality risk assessment in colorectal cancer patients. This nine-lncRNA prognostic signature is helpful to determine the individual mortality risk of overall survival and to improve the decision making of individualized treatments in colorectal cancer patients.

**Abbreviations**

| | |
|---|---|
| **CRC** | Colorectal cancer |
| **AUC** | Area under the ROC curve |
| **OS** | Overall survival |
| **ROC** | Receiver operating characteristic curve |
| **LncRNA** | Long non-coding RNA |
| **TCGA** | The Cancer Genome Atlas |
| **RPKM** | Reads per kilobase per million mapped reads |
| **TMM** | Trimmed Mean of M |
| **AJCC** | American Joint Committee on Cancer |
| **HR** | Hazard ratio |
| **CI** | Confidence interval |
| **GO** | gene ontology |
| **KEGG** | Kyoto Encyclopedia of Genes and Genomes |
| **DAVID** | Database for Annotation, Visualization, and Integrated Discovery |

### Funding

This study was supported by the Guangdong Provincial Health Department and the Guangdong Provincial Financial Department. The grant numbers were: B2018237 (grant recipient: Zhiqiao Zhang) and A2016450 (grant recipient: Zhiqiao Zhang). The total capital was RMB 15000. The funders had no role in study design, data collection and analysis, decision to publish, or preparation of the manuscript.

### Grant Disclosures

The following grant information was disclosed by the authors:
Guangdong Provincial Health Department: B2018237.
Guangdong Provincial Financial Department: A2016450.

### Competing Interests

The authors declare there are no competing interests.

### Author Contributions

- Zhiqiao Zhang conceived and designed the experiments, performed the experiments, analyzed the data, contributed reagents/materials/analysis tools, prepared figures and/or tables, authored or reviewed drafts of the paper, approved the final draft.
- Qingbo Liu conceived and designed the experiments, performed the experiments, analyzed the data, prepared figures and/or tables, authored or reviewed drafts of the paper, approved the final draft.
- Peng Wang conceived and designed the experiments, prepared figures and/or tables, authored or reviewed drafts of the paper, approved the final draft.
- Jing Li contributed reagents/materials/analysis tools, authored or reviewed drafts of the paper, approved the final draft.
- Tingshan He approved the final draft.
- Yanling Ouyang and Weidong Wang performed the experiments, analyzed the data, approved the final draft.
- Yiyan Huang contributed reagents/materials/analysis tools, approved the final draft.

### Human Ethics

The following information was supplied relating to ethical approvals (i.e., approving body and any reference numbers):

The data download and analyses were performed according to the policies of The Cancer Genome Atlas (TCGA) database. Since the study datasets in the present study were all downloaded from TCGA database, additional ethics approval was not needed. All data collections and analyses were in accordance with the principles of Declaration of Helsinki.

### Data Availability

All relevant data have been provided in the Supplemental Files.

## Supplemental Information

Supplemental information for this article can be found online at http://dx.doi.org/10.7717/peerj.6061#supplemental-information.

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
