# Peer review of "Development and internal validation of a nine-lncRNA prognostic signature for prediction of overall survival in colorectal cancer patients"

_PeerJ, doi:10.7717/peerj.6061_

## Round 0.1 · original submission · Minor Revisions

1. As colon and rectal cancer may have different prognostic factors, cancers in these two sites should be distinguished and analyzed (or at least described) separately.
2. The numbers for discovery and validation (see Table 1) need to be clear. How many of the 424 individuals were used for discovery and how many for validation? Please confirm that the numbers for these are accurate throughout the manuscript.
3. Change "wholly" in describing your lncRNA data to "total".
4. Include additional studies limitations in your discussion as relatively short follow-up time and few events. Also include text that the prognostic signature will need to be further tested in prospective studies.

Reviewer 1 ·

Basic reporting

no comment

Experimental design

no comment

Validity of the findings

no comment

Additional comments

This study is objected to developed and validated a nine-lncRNA prognostic signature for individual mortality risk assessment in colorectal cancer patients. The article has a good design framework. However, many similar articles have carried out the same research, so it seems that the innovation is insufficient.

Major comments:
1. Most of all is that the subject of article is about colorectal cancer, but the authors only used data on colon cancer of TCGA except rectal cancer. Can colon cancer represent all colorectal cancer?
2. According to Figure 1, 424 patients were included in final statistics, how did the authors divided these samples in model cohort (n=424) and validated cohort (n=424) in Table 1 ? The count here seems a bit contradictory.
3. There is limited overall survival (median 28.6 month follow up) and few events (108). For these analyses, it is very important to determine if these associations hold up in an additional cohort particularly one with longer follow up.

Minor comments:
Molecular mechanism of these nine-lncRNA prognostic signature could be discussed.

·

Basic reporting

Very clear and professional english through out the paper, I think. I am just unsure of whether "Wholly gene expression" is the correct wording, or "Whole" "Total" "Complete" would be more correct.

Experimental design

The research question is clear, and the attempt of developing and validating a prognostic model for colorectal cancer survival through a lnc-RNA signature is conducted to a high technical standard, and both methods and raw data is shared and described clearly.

Validity of the findings

no comment.

Additional comments

Very nice manuscript, within aims and scope for the peer journal. The mehod for developing this prognostic tool os interesting, and well described. I think this manuscript could be of interest for readers of peer journal.

---

## Round 0.2 · Major Revisions

As the validation set of samples is drawn from the same 424 individuals used to build the prognostic signature, this is a major concern. A second set of totally independent samples needs to be used for the validation studies.

The term "wholly" also needs to be changed in the abstract.

---

## Round 0.3 · accepted · Accept

Please do a final proofreading of the manuscript while in production.

#